# Pilot-Scale Production of Traditional Galotyri PDO Cheese from Boiled Ewes' Milk Fermented with the Aid of Greek Indigenous *Lactococcus lactis* subsp. *cremoris* Starter and *Lactiplantibacillus plantarum* Adjunct Strains

John Samelis [1,*], Charikleia Tsanasidou [1], Loulouda Bosnea [1], Charikleia Ntziadima [1], Ilias Gatzias [1], Athanasia Kakouri [1] and Dimitrios Pappas [2]

[1] Dairy Research Department, Institute of Technology of Agricultural Products, Hellenic Agricultural Organization 'DIMITRA', Katsikas, 45221 Ioannina, Greece; xaroulatsan@gmail.com (C.T.); louloudabosnea@gmail.com (L.B.); xntziadima@yahoo.gr (C.N.); iliasgr1985@yahoo.gr (I.G.); kakouriathanasia@yahoo.gr (A.K.)

[2] Skarfi EPE—Pappas Bros Traditional Dairy, 48200 Filippiada, Greece; pappasbr@otenet.gr

\* Correspondence: jsam@otenet.gr; Tel.: +30-2651094789

**Abstract:** The performance of a mixed thermophilic and mesophilic starter culture consisting of *Streptococcus thermophilus* ST1 and the Greek indigenous nisin-A-producing *Lactococcus lactis* subsp. *cremoris* M78 was evaluated in the absence (A: ST1+M78) or presence (B: ST1+M78+H25) of *Lactiplantibacillus plantarum* H25—another indigenous ripening strain—under real cheesemaking conditions. Three pilot-scale trials of fresh (6-day-old) Galotyri PDO cheese were made from boiled milk by an artisanal method using simple equipment, followed by cold ripening of the A1–A3 and B1–B3 cheeses at 4 °C for 30 days. All of the cheeses were analyzed microbiologically and for pH, gross composition, proteolysis, sugar and organic acid contents, and sensorial attributes before and after ripening. The artisanal (PDO) Galotyri manufacturing method did not ensure optimal growth of the ST1+M78 starter as regards the constant ability of the thermophilic strain ST1 to act as the primary milk acidifier under ambient (20–30 °C) fermentation conditions. Consequently, major trial-dependent microbial and biochemical differences between the Acheeses, and generally extended to the Bcheeses, were found. However, high-quality Galotyri was produced when either starter strain predominated in the fresh cheeses; only trial A1 had microbiological and sensory defects due to an outgrowth of post-thermal Gram-negative bacterial contaminants in the acidified curd. The H25 adjunct strain, which grew above 7 to 9 log CFU/g depending on the trial, had minor effects on the cheese's pH, gross composition, and proteolysis, but it improved the texture, flavor, and the bacteriological quality of the Bcheeses during processing, and it exerted antifungal effects in the ripened cheeses.

**Keywords:** Galotyri PDO cheese; *Lc. lactis* subsp. *cremoris* M78; *L. plantarum* H25; acid-curd cheese





## 1. Introduction

In recent years, most traditional Greek cheeses have been produced from pasteurized or thermized ewes' or goats' milks, or their mixtures, in order to eliminate product losses and safety risks associated with use of raw milk [1–4]. Hence, the application of efficient and well-standardized lactic acid bacteria (LAB) starter cultures is required for promoting optimal rates of acidification and fermentation of the heat-treated cheese milks [5,6]. Direct vat-set (DVS) commercial starter cultures (CSCs) consisting of defined multistrain mixtures of *Streptococcus thermophilus*, *Lactobacillus helveticus*, *Lactobacillus delbrueckii* (of the subspecies *delbrueckii*, *bulgaricus*, and *lactis*), *Lactococcus lactis* (of the subspecies *lactis*, *cremoris*, and the variant *diacetylactis*), and *Leuconostocmesenteroides* subsp. *cremoris* are preferable for stabilizing commercial cheesemaking technologies and ensuring cheese safety [5–7]. Depending on the cheese category, CSCs may contain solely thermophilic, solely mesophilic

or, most frequently, mixed thermophilic and mesophilic LAB strains [6,8], which are either natural isolates of dairy origin or selected original isolates that have been genetically manipulated for desirable biotechnological traits [9–11].

Although Greece has a very long tradition in the production and consumption of cheese [3,12], all CSCs currently available in the Greek market are imported; most of them are stable large-scale products of international CSC manufacturers, distributed mainly in freeze-dried form and applied routinely in the dairy industry and most traditional dairies [7,13,14]. Imported DVS CSCs have become widespread in Greece because their selection and application is much easier and far more convenient than the production and application of undefined, natural raw milk, whey, or yogurt-like starter cultures that are difficult to preserve and handle in commercial dairy plants [5,9,15]. However, apart from their advantages, the constant use of 'foreign' CSCs in traditional Greek cheese technologies has important disadvantages [16]. In general, CSCs include natural [9] or generated [10,11] dairy LAB strains that are not indigenous to Greece, and whose actual performance in milk types other than cow milk has not necessarily been evaluated. Moreover, the LAB strain constituents in CSC packages are mixed at constant ratios that are specified by the CSC manufacturer and cannot be altered; only their total inoculation level in the cheese milk vat can be adjusted. Moreover, DVS CSCs are applied at quite high ($\geq$7 log CFU/mL) total cell inoculation levels; thus, the prevalent growth of industrial starter LAB strains negatively influences biodiversity in traditional Greek cheeses by eliminating or suppressing the growth of indigenous LAB, including enterococci, *Leuconostoc*-like bacteria, mesophilic lactobacilli, and wild lactococci, as well as coagulase-negative staphylococci and other non-LAB ripening biota [2,9,16]. It is well known that nonstarter LAB persisting in milk and traditional dairies exert beneficial biotechnological activities and produce bacteriocins and other natural antimicrobials in situ in artisanal cheese fermentations [17–19]. Bacteriocin-producing (Bac+) LAB strains are not included as adjuncts in multistrain CSC products, because they may negatively influence the growth and overall performance of the primary starter LAB strains. Lastly, the exclusive use of imported CSCs in traditional Greek protected designation of origin (PDO) cheese production is considered to be an important biotechnological limitation, despite it being permitted by the national legislation and food regulatory authorities [20]. The fact that the same imported CSC types are distributed and may be applied for the manufacture of the same or similar traditional cheese types in different dairy plants all over Greece is contrary to the preservation of cheeses' biodiversity and authenticity—particularly of the most popular and important Greek cheese varieties certified with PDO recognition [4,16,21].

Thus, the development of novel dairy starter culture preparations of improved performance in traditional Greek cheese technologies is required [22]. Recent studies in the framework of the national project BIOTRUST [23] first searched for new indigenous Greek LAB strains and mapped their key technological and functional characteristics [24,25]. Next, we searched for compatible combinations of indigenous bacteriocinogenic Greek strains, such as the novel nisin-A-producing *Lc. lactis* subsp. *cremoris* M78/M104 strain genotype and the enterocin-A-B-P-producing strain *Enterococcus faecium* KE82, with natural or primary CSC strains of *S. thermophilus*, in milk model experiments [26,27] or under real factory-scale cheese manufacturing conditions. For the latter task, Galotyri PDO—the oldest and most popular traditional Greek acid-curd cheese [2–4,12]—served as the most suitable cheese model for evaluating the growth compatibility and performance of newly designed starter/adjunct LAB strain combinations [28]. Historically, the authentic Galotyri is produced in the regions of Epirus and Thessaly from boiled ewes'/goats' milk, which is cooled in clay jars for 24 h and then salted with 3–4% edible salt, before being left to acidify and curdle naturally, with or without rennet, at ambient temperature for 2–3 days, followed by cold ripening ($\leq$8 °C) of the fresh cheese in leather bags or wooden barrels for at least two months [20]. In the past, the natural acidification of the boiled milk required for Galotyri curd formation was ensured by a diverse adventitious LAB 'house-flora' or the use of empirical back-slope techniques for milk inoculation. At present, however, all industrial

and most artisan-type Galotyri PDO cheese products are manufactured from pasteurized milk fermented with the aid of various CSCs or natural yogurt-like cultures [4,16], and they are also marketed fresh or ripened for only a short time to increase profit [29]. In some traditional dairies, the lack or shortening of ripening has been replaced by the dispersion of ripened Feta or whey cheese granules in the basal yogurt-like Galotyri curds to enhance the aromatization and taste of the fresh retail cheese [16,29]. However, this arbitrary practice, used as an alternative to ripening, should be neither permitted nor encouraged, because it is not specified in the Galotyri PDO technology [20]. Instead, traditional cheese processors in Epirus and Thessaly must be assisted by dairy researchers in developing, preserving, and applying simple or more diversified consortia of novel indigenous starter or adjunct LAB strains in the boiled or pasteurized milk intended for real Galotyri cheese production.

This study aimed to evaluate the main quality attributes of Galotyri PDO cheese manufactured with an artisanal cheesemaking protocol. Pilot-scale Galotyri cheese trials from boiled ewes' milk fermented with an *S. thermophilus* natural starter, combined with two prime Greek indigenous *Lc. lactis* and *L. plantarum* strains, were performed. In particular, a new promising indigenous *L. plantarum* strain was evaluated as an adjunct ripening culture in the Galotyri trials of this study, concurrent with the fact that *L. plantarum* has been found to be the most prevalent nonstarter LAB species in several retail Galotyri PDO or Galotyri-like cheese varieties [4,16,29,30]. The ready-to-eat (RTE), fresh or ripened cheese products were compared on the basis of their microbiological, physicochemical, biochemical, and sensorial characteristics.

## 2. Materials and Methods

### 2.1. Bacterial Strains, Starter/Adjunct LAB Strain Combinations, and Culture Conditions

Two indigenous LAB strains—the wild nisin-A-producing (NisA+) strain *Lc. lactis* subsp. *cremoris* M78, originally isolated from Greek raw mixed ewes'/goats' (90:10) milk [1]; and the nonstarter ripening strain *L. plantarum* H25, originally isolated from traditional Greek Graviera cheese [13]—were selected for use in this study. Selection was based on data from previous biochemical and molecular characterizations [24,31–34] and factory-scale cheesemaking [35] studies, which altogether revealed that strains M78 and H25 are safe and possess high in situ biotechnological, antilisterial and/or probiotic potential in traditional cheese milk fermentations.

For the purposes of this study, each of the strains (M78 and H25) was combined with *S. thermophilus* CSL-ST1—a genetically unmodified starter strain of natural origin derived from the blend of a CSC (GRU IDC 01, Centro Sperimentale del Latte (CSL), Lodi, Italy), hereafter referred to as ST1—and used as the primary starter LAB species for the acidification of 'boiled' milk, as described in the findings of Samelis and Kakouri [26]. Additionally, an overall good growth compatibility and sufficient nisin A gene expression and production were found previously in sterile raw milk (SRM) co-cultures of the strain combinations ST1+M78 (i.e., the basic starter formula) and ST1+M78+H25 (i.e., the basic starter supplemented with the H25 strain as an adjunct) during SRM fermentation at 37 °C for 6 h, followed by an additional 66 h at 22 °C [27]. Accordingly, the application of the above two starter/adjunct LAB strain combinations was extended to the artisanal pilot-scale Galotyri cheese production trials of this study, as described in Section 2.2 below.

Strains were sub-cultured twice for reactivation: M78 and H25 in MRS broth (Neogen Culture Media, formerly Lab M, Heywood, UK) incubated at 30 °C for 24 h, and *S. thermophilus* ST1 in M17 broth (Merck, Darmstadt, Germany) incubated at 37 °C for 24 h. For preparation of the Galotyri cheese milk inocula, all strains were cultured in 10 mL portions of heat-sterilized (121 °C, 5 min), 10% reconstituted skimmed milk (RSM) powder (Lab M), incubated at the optimal growth temperature of each strain for 24 h.

### 2.2. Artisanal Galotyri Cheese Preparation and Sampling

Three individual cheese trials were processed following the traditional PDO Galotyri manufacturing method, with boiled milk and simple artisanal practices and equipment,

as described by Sameli et al. [28]. The first and third cheese trials were conducted in a traditional dairy plant (Pappas Bros., Skarfi E.P.E., Filippiada, Epirus, Greece) in December 2019 and April 2021, respectively. The second, intermediate cheese trial was conducted in February 2020 in the pilot plant of the Dairy Research Department (Ioannina, Epirus, Greece) to increase supervision of the hygienic precautions during processing. Supervision was required because high levels of accidental contamination with Gram-negative spoilage bacteria occurred during cheese milk fermentation in the first trial (see Section 3), which prompted us to resolve this problem prior to conducting additional cheese trials at the Pappas plant. The artisanal Galotyri manufacturing protocol and the cheese milk/curd sampling protocol during processing are summarized in Figure 1 and detailed below.

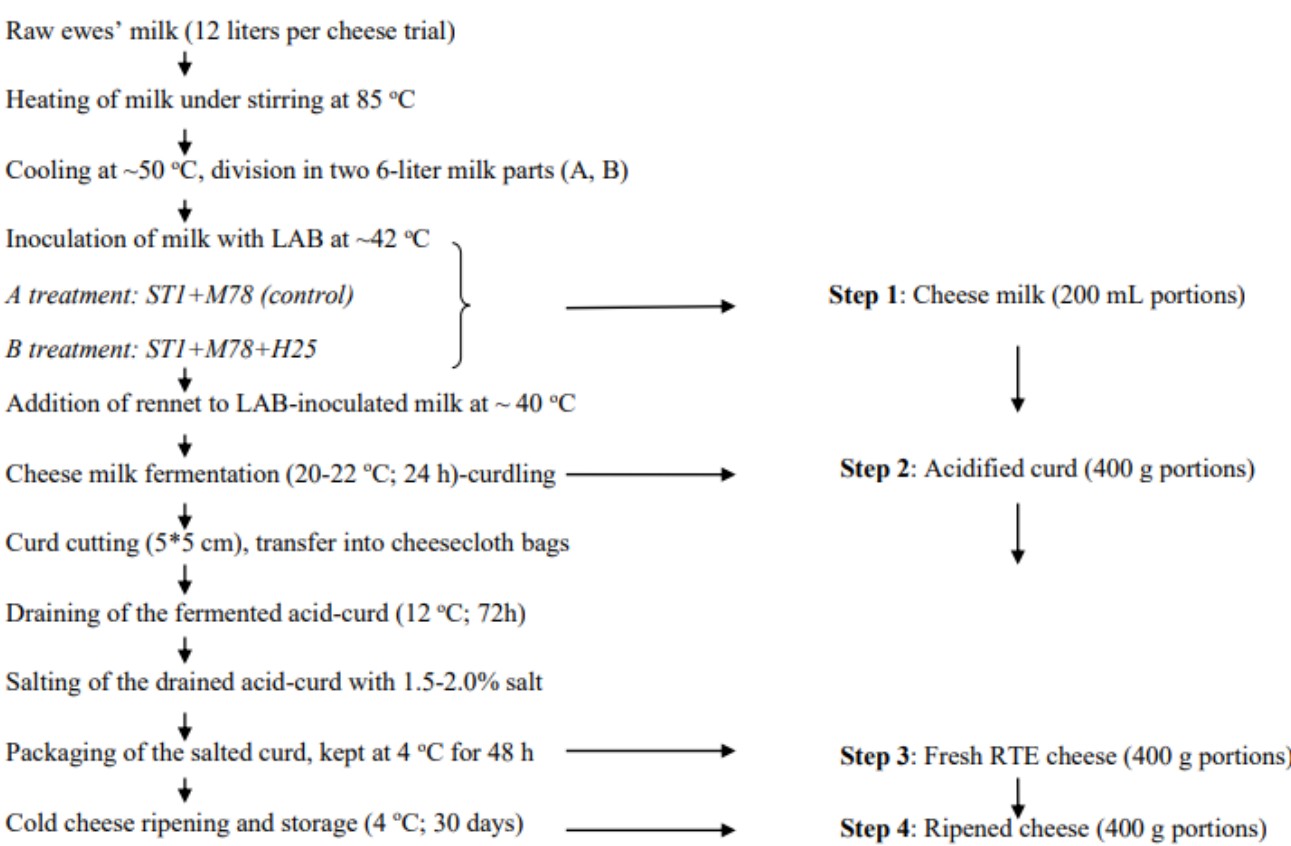

**Figure 1.** Artisanal Galotyri PDO cheese processing and sampling protocol.

For each experimental cheese trial, 12 L of raw ewes' milk obtained from local farmers was heated slowly in a stainless steel vessel at 85 °C over a gas burner under continuous stirring. The temperature was recorded with a probe thermometer. The milk vessel was then placed into a basin with cold running water to accelerate cooling. When the temperature fell to 50 °C, the milk was divided in two 6 L portions in two small (10 kg) plastic (trial 1) or stainless steel (trials 2 and 3) buckets with lids. At 42 °C, the milk was inoculated with appropriate volumes of fresh (24 h) LAB cultures in RSM to obtain approximate inoculation levels of 6.5 to 7.0 log CFU/mL for *S. thermophilus* ST1, 6.0 to 6.5 log CFU/mL for *Lc. lactis* subsp. *cremoris* M78, and 5.5 to 6.0 log CFU/mL for *L. plantarum* H25. In the first bucket, strains ST1 and M78 were added as the basic starter culture (ST1+M78; Galotyri A/control), whereas in the second bucket the H25 adjunct strain was added after inoculation with the basic starter ST1+M78 culture (Galotyri B; new product). Then, each of the three Galotyri A batch trials inoculated with the basic starter ST1+M78 (A1, A2, and A3) was processed further, sampled, analyzed in parallel, and compared with each of the corresponding Galotyri B batch trials (B1, B2, and B3) containing the ST1+M78+H25 combination, as follows: The LAB-inoculated cheese milks were stirred (step 1; termed

cheese milk), the first samples (200 mL) were collected, and then 100 μL of liquid rennet (80% chymocin, 20% pepsin; strength 1:15.000, Semi Piccante 80PS15, Bioren, Corato, Italy) was added to each cheese milk at ca. 40 °C to enhance curdling. Lids were added to the buckets, and the cheese milks were left to curdle in each plant's fermentation chamber at 20 to 22 °C for 24 h (step 2; termed acidified curd). The time required for the milk temperature to decrease from 40 °C after the addition of rennet to 20–22 °C during fermentation could not be monitored. During the next day, the fresh cheese curds were cut into large cubes (ca. 5 by 5 cm) with a flamed stainless steel knife, and then transferred to a thick cheesecloth bag for draining in a cooling chamber at 12 °C for 72 h. After draining, the curds were transferred to clean 5 kg plastic containers with lids, and 1.5 to 2.0% granular edible sea salt was added to each curd mass gradually, followed by hand kneading to facilitate salt distribution. After salting, the containers were transferred to each plant's refrigerator (4 °C) for 48 h to enhance salt diffusion. This salted 6-day-old acid-curd product (step 3) was defined as the fresh RTE Galotyri cheese. All cheeses were covered with a food-wrapping film membrane to prevent mold growth and were ripened at 4 °C for 30 days in their plastic containers (step 4; termed ripened cheese).

All Galotyri cheese trials (A1 to A3 and B1 to B3) were analyzed microbiologically and for pH after the four sequential processing steps 1 to 4 specified above (Figure 1). Additionally, the fresh RTE (step 3) and ripened (step 4) Galotyri cheese products were subjected to physicochemical and sensorial analyses. All cheese milk (200 mL) and curd (400 g) samples were collected aseptically for microbiological analyses, pH and moisture determination, and sensory evaluation on the day of sampling, or they were divided into portions and stored in a freezer (−30 °C) for later biochemical analyses.

*2.3. Microbiological Analyses of the Artisanal Galotyri Cheese Samples*

On each sampling occasion, 25 g cheese milk or curd samples were homogenized with 225 mL of sterile quarter-strength Ringer solution (Lab M) in stomacher bags (Lab Blender, Seward, London, UK) for 60 sec at room temperature. Then, 1 mL portions of the homogenates were decimally diluted in tubes with 9 mL of sterile quarter-strength Ringer solution, and appropriate dilutions were poured (1 mL samples) or spread (0.1 mL samples) in duplicate on total and selective agar media plates. Unless stated otherwise, all agar media and their supplements were purchased from Neogen Culture Media (formerly Lab M, Heywood, UK).

Total viable counts (TVCs), including the mixed populations of the ST1, M78, and H25 strains, were enumerated on Milk Plate Count Agar (MPCA) incubated at 37 °C for 48 h. Thermophilic dairy LAB populations were enumerated on M17 agar incubated at 45 °C for 48 h. This analysis was simultaneously used for the selective enumeration of the primary starter strain *S. thermophilus* ST1, because the other two mesophilic LAB strains (M78 and H25) cannot grow at 45 °C, while the contamination and growth of autochthonous LAB—mainly enterococci and thermophilic lactobacilli, able to grow at 45 °C—in the 'boiled' Galotyri cheese milk were expected to be at much lower levels than the ST1 strain's population density. Moreover, in this study, enterococci—which generally form larger colonies than the ST1 colonies on M17 agar at 45 °C—were selectively enumerated in parallel on Slanetz and Bartley (SB) agar, incubated at 37 °C for 48 h. ST1 and M78 cannot grow on SB. On the other hand, H25 promoted unhindered growth in the form of white (colorless) colonies; however, these were easily discriminated from the reddish-brown *Enterococcus* colonies on the SB agar plates. Comparative enumerations of total mesophilic LAB populations were made on MRS agar incubated at 30 °C for 48–72 h, and on M17 agar incubated at 22 °C for 72 h. Specifically, accurate selective enumerations of *L. plantarum* H25 populations in the Galotyri B (ST1+M78+H25) cheese trials were made in the form of its very characteristic, yellowish, large and convex colony growth on MRS agar plates after 48–72 h of incubation at 30 °C. On the other hand, selective enumeration of the M78 colony populations was facilitated macroscopically on the M17/22 °C agar plates, because the ST1 and H25 strains displayed weak (i.e., pinpoint) colony growth on M17 agar at

22 °C after 48 to even 72 h of incubation. Moreover, accurate selective enumerations of the NisA+ strain M78 colonies were obtained by pouring the surface of high-dilution M17/22 °C and MPCA/37 °C agar plates of all tested samples with fresh (24 h) cell lawn suspensions of *Listeria monocytogenes* no.10 in melted (45 °C) tryptic soy agar with 0.6% yeast extract (TSAYE)—a simple and effective procedure known as the agar overlay assay [13]. After solidification, the agar plates were incubated at 30 °C overnight. On the next day, large listerial inhibition zones surrounded the NisA+ M78 colonies, if present on the agar plates, whereas the ST1 and H25 colonies were not inhibitory when grown on M17 or MPCA agar media. The reliability of the agar overlay assay was confirmed by pouring fresh *L. monocytogenes* no. 10 lawns onto the M17/45 °C agar plates, which supported ST1 colony growth but were free of M78 colony growth due to the high incubation temperature; thus, with very few exceptions (to be discussed in later paragraphs), all colonies grown on the M17/45 °C agar plates of all Galotyri cheese samples did not inhibit *Listeria* growth.

Additionally, total staphylococci were enumerated on a Baird-Parker agar base with egg yolk tellurite (BP), incubated at 37 °C for 48 h; pseudomonad-like bacteria were enumerated on a *Pseudomonas* agar base supplemented with cetrimide–fucidin–cephaloridine (CFC), incubated at 25 °C for 48 h; coliform bacteria were enumerated by pouring 1 mL samples into melted (45 °C) violet red bile (VRB) agar, overlaid with 5 mL of the same medium and incubated at 37 °C for 24 h; yeasts and molds were enumerated on Rose Bengal Chloramphenicol (RBC) agar (Merck, Darmstadt, Germany), incubated at 25 °C for 5 days.

### 2.4. Measurement of pH and Gross Composition of the Artisanal Galotyri Cheese Samples

The pH of all Galotyri cheese milk or curd samples was measured with a Jenway 3510 digital pH meter (Essex, UK) after plating for microbiological analysis. All fresh RTE and ripened Galotyri cheese samples were analyzed for moisture, fat, protein, salt, ash, and titratable acidity. Moisture was measured by drying to constant weight at $102 \pm 1$ °C [36]. Fat content was determined according to the Gerber method [37], protein content by the Kjeldahl method [38], and salt content by the modified Volhard method [39]. Ash content was measured by dry ashingin afurnace at 550 °C, following the method described by the Association of the Official Analytical Chemists (AOAC) standard 935.42 [40]. Total titratable acidity was measured by the Dornic method after mixing 10 g of cheese with an equal mass of distilled water and expressed as % (*w/w*) lactic acid [41].

### 2.5. Assessment of Proteolysis in the Artisanal Galotyri Cheese Samples

Cheese proteolysis was assessed by measuring different nitrogen fractions with established methods, as previously described by Mallatou et al. [42]. Briefly, the total nitrogen (TN) content was estimated by the Kjeldahl method. Water-soluble nitrogen (WSN) and nitrogen soluble in 12% trichloroacetic acid (TCA-SN) were determined in samples of extracts, as described by Kuchroo and Fox [43]. The Sorval Omni-Mixer (DuPont Company, Newton, CT, USA) was used for homogenization, and the supernatant was filtered through no.42 filter paper. Nitrogen soluble in 5% phosphotungstic acid (PTA-SN) was also determined by the Kjeldahl method as described by Stadhouders [44], with the exception that the extract was prepared as mentioned above. To assess proteolysis, measurement of all three nitrogen fractions was required, because (i) the WSN fraction consists of all proteolysis products and of the whey proteins that have been retained in the curd, (ii) the TCA-SN fraction includes free amino acids and small–medium-sized peptides with 2–20 amino acid residues, and (iii) the PTA-SN fraction includes free amino acids and very small peptides [45]. From the TN and the three nitrogen fraction measurements, the respective ripening indices were calculated as follows: ripening extension index (REI) = WSN/TN, ripening depth index (RDI) = TCA/TN, and free amino acid index (FAAI) = PTA/TN [46].

## 2.6. Determination of Sugar and Organic Acid Concentrations

The concentrations of the main sugars D-lactose, D-glucose, and D-galactose in Galotyri cheese were determined with the use of the respective enzymatic kits of R-Biopharm, Boehringer Mannheim (Darmstadt, Germany). Cheese samples were extracted separately according to the proposed methods of the enzymatic kits for each determination.

The reagents used for determination of the concentrations of the main organic acids in Galotyri cheese by HPLC were lactic acid 85%, citric acid, and phosphoric acid ($H_3PO_4$) 85% obtained from Mallinckrodt Chemical Works (Chicago, IL, USA), acetic acid 100% obtained from Merck (Darmstadt, Germany), and propionic acid 99% obtained from Sigma Chemical Works (Burlington, VT, USA); ultrapure water (UPW) prepared using a Milli-Q water purification (Millipore Simplicity UV, Merck, Darmstadt, Germany) was used in the reagent preparations.

Standard solutions of each organic acid were prepared by dissolving the individual acid standard in an appropriate volume of UPW to obtain concentrations of 1000–10,000 mg/L. Standard mixtures were prepared by mixing 1 mL of each standard solution. Determination of organic acids in the samples was carried out by extraction using the method of Mortera et al. [47]. One gram of each cheese sample was diluted to 10 mL with water and blended with a vortex. Then, the samples were centrifuged at $3000 \times g$ for 15 min.

The extracts (2 µL) from the WSN fraction were filtered through a 0.22 µm filter and analyzed on an LC-20AT high-performance liquid chromatograph (Shimadzu, Tokyo, Japan) equipped with a thermostatted autosampler (SIL-20A), a high-pressure binary mixing pump (LC-20AT), a column oven (CTO-20AC), and a diode array detector (SPD-M20A). Separation of the extracts was carried out on an YMC-Triart C18 column (3 µm, 150 × 3 mm I.D., Shimadzu, Kyoto, Japan). Organic acids were separated using an isocratic elution program with $H_3PO_4$ 0.02 M as the mobile phase at a flow rate of 0.425 mL/min. The column temperature was set to 37 °C, and the peaks were detected at 220 nm. Identification of the organic acids was based on their retention times according to a standard organic acids curve.

## 2.7. Sensory Evaluation

Artisanal RTE Galotyri cheese products manufactured without (ST1+M78) or with the *L. plantarum* H25 adjunct strain (ST1+M78+H25) were compared organoleptically after 15 days of ripening at 4 °C only, because multiple sensory evaluation panel meetings were prevented by the sequential working restrictions applied due to the COVID-19 pandemic. Sensory evaluation was conducted by a trained five-member panel. All of the panelists belonged to the permanent staff of the Dairy Research Department and were familiar with traditional acid-curd cheeses. Sensory evaluation was performed as described by Kondyli et al. [48]. The panelists were asked to evaluate the cheeses for appearance, body and texture, and flavor (odor and taste) using a 10-point scale, with 1 being the worst and 10 the best quality. Importance was given predominantly to the attributes of flavor and of body and texture, rather than the appearance of the cheese. Thus, the grade scores obtained for these two attributes were multiplied by 5 and 4, respectively. The total score was obtained by adding the scores for the three sensory attributes. An excellent cheese received a total score of 100. The panelists used water to clean their mouths between samples; they were also instructed to report the defects foreach set of attributes, according to the cheese defect descriptions and general guidance given by the IDF reference method [49].

## 2.8. Statistical Analysis

Three independent pilot-scale Galotyri cheese trials were analyzed (*n* = 3). Within each trial, the values reported for each microbiological and physicochemical parameter were the means of two individual measurements. The microbiological data were converted to log CFU/g and, along with the data for the physicochemical and sensory parameters, were subjected to a one-way analysis of variance using the software Statgraphics Plus for Windows v. 5.2 (Manugistics, Inc., Rockville, MD, USA). The means were separated by

the LSD procedure at the 95% confidence level ($p < 0.05$) for determining the significance of differences in each Galotyri cheese treatment with time, and between the two Galotyri cheese treatments A and B at each sampling step.

## 3. Results

### 3.1. Growth of Starter LAB and Native Microbiota in the Artisanal Galotyri Cheeses

The growth patterns of total (starter and native) microbiota (TVC; MPCA/37 °C), total thermophilic dairy (lactose-fermenting) LAB (M17/45 °C), total mesophilic dairy LAB (M17/22 °C), total mesophilic LAB (MRS/30 °C), enterococci (SB/37 °C), and yeasts (RBC/25 °C) across four critical processing and ripening steps of the artisanal Galotyri A (ST1+M78) and Galotyri B (ST1+M78+H25) cheese samples are shown in Figure 2A,B, respectively.

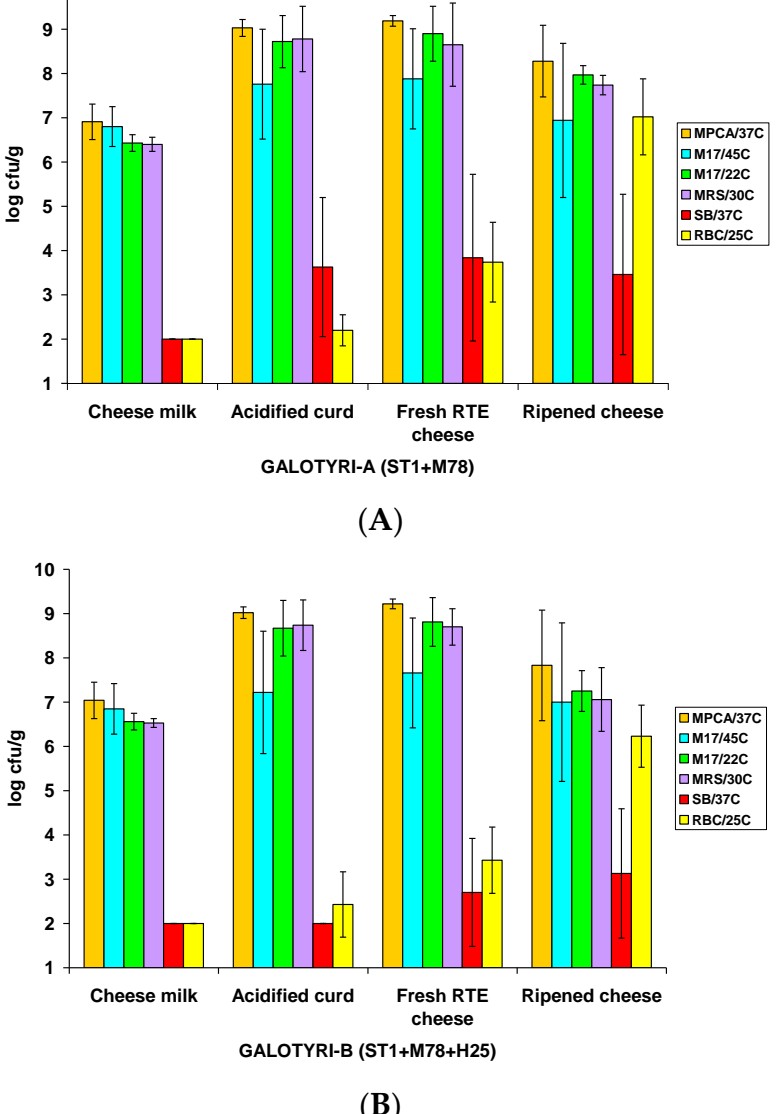

**Figure 2.** Microbial evolution in artisanalGalotyri cheeses manufactured without (**A**) or with (**B**) supplementation of the basic starter culture (ST1+M78) with the adjunct strain *L. plantarum* H25.

The mean total viable counts (TVCs), mainly reflecting the total dominant starter/adjunct LAB populations in boiled (85 °C) milk after inoculation with the ST1, M78, and H25 strains (step 1; Figure 1), were 6.91 ± 0.40 log CFU/g and 7.04 ± 0.41 log CFU/g in the Galotyri A and Galotyri B cheese milk samples, respectively. The TVCs increased by ca. 2-log units ($p < 0.05$) during cheese milk fermentation and reached similarly high levels of

9.03 ± 0.19 log CFU/g and 9.02 ± 0.13 log CFU/g in the acidified curds of the Galotyri A (Figure 2A) and Galotyri B (Figure 2B) cheeses, respectively. From the acidified curd (step 2; Figure 1) to the fresh RTE cheese (step 3; Figure 1), slight further TVC increases ($p > 0.05$) occurred on average, followed by overall significant ($p < 0.05$) TVC decreases after 30 days of cold ripening (step 4; Figure 1). Therefore, the ripened Galotyri A and Galotyri B cheeses had reduced TVC levels, which were 8.28 ± 0.81 log CFU/g and 7.83 ± 1.25 log CFU/g, respectively (Figure 2A,B).

The mean total mesophilic LAB populations grown on either MRS/30 °C or M17/22 °C agar plates predominated in the acidified curds (step 2) and continued to be highly predominant in the fresh RTE samples (step 3) of both Galotyri cheese types (Figure 2). In contrast, the growth of total thermophilic dairy LAB was more restricted (<8 log CFU/g; $p < 0.05$) than that of total mesophilic LAB during cheese milk fermentation. Thus, their mean populations in the fresh RTE Galotyri A and Galotyri B cheese samples were 7.88 ± 1.13 log CFU/g (Figure 2A) and 7.66 ± 1.24 log CFU/g (Figure 2B), respectively—a result that actually indicated the failure of the primary starter *S. thermophilus* ST1 to predominate in all acidified Galotyri curds. Thermophilic and mainly mesophilic LAB populations declined in the ripened cheeses, in accordance with the TVC declines (Figure 2). Autochthonous enterococci were <100 CFU/g in all 'boiled' cheese milks, but afterwards their populations fluctuated greatly (from <2.00 to 5.96 log CFU/g) among the three pilot-scale Galotyri cheese trials during processing. Generally, enterococci reached higher ($p < 0.05$) mean populations in the fresh (mainly) and the ripened Galotyri A (Figure 2A) than in the corresponding Galotyri B (Figure 2B) cheese samples. Finally, yeasts increased greatly (>2.5 to 3.5 log units) in all cheese samples during ripening;their final populations were 7.02 ± 0.86 log CFU/g and 6.23 ± 0.70 log CFU/g in the ripened Galotyri A and Galotyri B samples, respectively. Thus, in general, the growth of enterococci and yeasts was retarded in the presence of the *L. plantarum* H25 adjunct strain during the processing and ripening of the Galotyri B cheese samples (Figure 2).

Selective starter/adjunct colony enumerations are essentially presented separately for each Galotyri cheese trial to demonstrate the strong trial-dependent effects on the growth patterns of the ST1, M78, and H25 strains noted during this study (Table 1). Indeed, contrary to our expectations, the primary starter strain *S. thermophilus* ST1 failed to promote growth in both treatments of the two trials (A1/B1 and A3/B3) conducted in the commercial plant (Table 1). Particularly, in the first trial, the inoculated (>7 log CFU/g) ST1 populations in cheese milk declined below 6 log CFU/g in the acidified Galotyri A and -B curds fermented at ambient temperature for 24 h, according to the artisanal manufacturing method; then, ST1 tended to recover and grow slightly in the fresh A1 cheese (7.67 ± 0.10 log CFU/g), but not in the fresh B1 cheese, where it declined below 5 log CFU/g (Table 1). In the third trial, inoculated (ca. 7.0 log CFU/g) ST1 populations did not decline but failed to grow (<0.3 log CFU/g increase) in both treatments; thus, they failed to prevail in the respective acidified curd and fresh RTE cheese (A3 and B3) samples. Consequently, ST1 occurred at low (5.4–6.5 log CFU/g) population levels in the ripened cheeses of both trials conducted in the commercial plant. In starkcontrast, ST1 grew predominantly in the acidified curds, exceeding 9 log CFU/g in the fresh Galotyri A2 and B2 cheese samples of the second trial conducted in the DRD pilot plant under supervision, and remained at this high population level in the ripened Galotyri A2 and B2 cheeses (Table 1).

Growth of the NisA+ mesophilic co-starter strain *Lc. lactis* subsp. *cremoris* M78 was significantly affected by the growth pattern of *S. thermophilus* ST1 during artisanal Galotyri cheese processing. Specifically, M78 promoted good growth (ca. 1.5 log CFU/g increase; $p < 0.05$) in the acidified curd and the fresh A2 and B2 cheese samples of the second trial, but it remained subdominant (7.9–8.2 log CFU/g) compared to the predominant primary starter ST1. In contrast, M78 predominated at population levels of 8.97 to 9.34 log CFU/g in the absence of ST1 growth in the acidified curd and the fresh RTE cheese samples of the first and third Galotyri trials conducted in the commercial plant (Table 1). A clear exception was the control A1 treatment of the first trial, where both ST1 and M78 strains failed to

grow in the curd and afterward to exceed 8 log CFU/g in the fresh RTE cheese (Table 1). M78 populations declined ($p < 0.05$) in the ripened cheeses of all trials. However, the loss of M78 viability was variable; it was the highest in the second trial, dominated by ST1 during processing and ripening. Particularly, in the ripened A2 cheese, the final M78 populations declined below 4 log CFU/g, accounting for a greater than 4-log reduction during ripening. This finding was indicative of strong autolysis of the NisA+ M78 cells during A2 cheese ripening. Conversely, the M78 populations declined the least (1.27 log CFU/g reduction) from the fresh to the ripened A1 cheese steps (Table 1).

The adjunct strain *L. plantarum* H25 grew significantly ($p < 0.05$) in all acidified curd samples after the first 24 h of cheese milk fermentation. However, its growth was variable; it was the highest (2.77 log CFU/g increase) in the B3 curds, where it reached an almost equal level of dominance with strain M78, whereas it was the lowest (1.0 log CFU/g increase) in the B2 curds, where the primary starter strain ST1 predominated. Afterwards, the H25 populations either declined (B3; ca. 0.7 log; $p < 0.05$), remained stable (B1; $p > 0.05$), or increased further (B2; ca. 0.7 log; $p < 0.05$) in the fresh RTE cheese samples. In terms of its potential probiotic use, strain H25 remained viable at ca. 7 log CFU/g or higher levels in all fresh RTE Galotyri cheeses, as well as in the ripened B2 and B3 cheeses, as is desirable for probiotic cultures. Only the ripened B1 cheese had fairly low (<6.5 log CFU/g) H25 populations (Table 1).

Overall, the control A1 (ST1+M78) treatment of the first Galotyri trial was a problematic artisanal cheese production because of an accidental high contamination and growth of Gram-negative spoilage bacteria during the critical cheese milk fermentation step (Table 2). Specifically, coliform and pseudomonad-like bacteria predominated at levels as high as $7.80 \pm 0.04$ log CFU/g and $7.51 \pm 0.04$ log CFU/g in the A1 acidified curds, which also displayed the highest contamination and growth of autochthonous enterococci and staphylococci among all curds. After 6 days, Galotyri A1 was the only fresh RTE cheese that retained high contamination levels (>6.0 to 7 log CFU/g) of Gram-negative spoilage bacteria and a quite high (ca. 6.0 log CFU/g) level of enterococci. Conversely, the growth of Gram-negative bacteria and enterococci was significantly reduced in the curd and the fresh RTE Galotyri B1 cheese (Table 2) by the predominant growth of the NisA+ strain M78 and sufficient co-growth of the adjunct strain H25 (Table 1). The growth of enterococci and non-LAB biota was also prevented or reduced significantly in the later two artisan Galotyri cheese trials. Particularly, the second trial, conducted at the DRD pilot plant under strict hygienic conditions, contained less than 100 CFU/g of enterococci, staphylococci, and Gram-negative bacteria throughout processing and ripening. Pseudomonad-like spoilage bacteria were fully suppressed during the processing and ripening of the Galotyri A3 and B3 cheeses in the commercial plant. However, the occurrence of few (<4 log CFU/g) enterococci, staphylococci, and coliforms in the acidified curd and the fresh and/or ripened Galotyri A3 samples was unavoidable; their levels were minimized ($p < 0.05$) in the respective Galotyri B3 samples, confirming an increased growth inhibition of enterococci and non-LAB biota by the M78+H25 strain combination (Table 2).

As mentioned, yeasts attainedmajor ($p < 0.05$) growth increases, ranging from 2.52 to 3.65 log CFU/g, in all Galotyri trials during ripening (Table 2). Notably, the fresh and ripened Galotyri A2 and B2 cheeses—which werethe only trials highly dominated by ST1—displayed the highest yeast population levels among all trials. Moreover, in all trials, the final yeast growth was higher in the ripened Galotyri A than in the Galotyri B cheese samples (Table 2), confirming that the adjunct strain *L. plantarum* H25 had antifungal effects (Figure 2). Another useful finding was that the dominant yeast type(s) in the ripened Galotyri A1 and B1 cheeses, selectively counted on the RBC agar plates, were also able to grow well as small whitish colonies on the CFC agar plates (Table 2)—a medium that is considered to be highly selective for the enumeration of *Pseudomonas* species and Gram-negative bacteria overall.

**Table 1.** Populations (log CFU/g) of the primary starter *S. thermophilus* ST1, the NisA+ co-starter *Lc. lactis* subsp. *cremoris* M78, and the adjunct *L. plantarum* H25 strains selectively enumerated infour sequential processing steps of three independent batches of pilot-scale-produced, traditional Galotyri PDO cheese [1].

| Starter or Adjunct LAB Strain | | Galotyri Cheese A (Basic Starter Mix: ST1+M78) [2] | | | | | Galotyri Cheese B (Starter Mix: ST1+M78+H25) [2] | | | |
|---|---|---|---|---|---|---|---|---|---|---|
| | Trial/ Batch | Cheese Milk | Acidified Curd | Fresh RTE Cheese | Ripened Cheese | Trial/ Batch | Cheese Milk | Acidified Curd | Fresh RTE Cheese | Ripened Cheese |
| *Streptococcus thermophilus* ST1 | | | | | | | | | | |
| | A1 | 7.19 ± 0.06 cA | <6.00 bA | 7.67 ± 0.10 cB | 5.87 ± 0.02 bA | B1 | 7.39 ± 0.04 cB | <6.00 bA | <5.00 aA | 5.41 ± 0.15 bA |
| | A2 | 6.36 ± 0.06 aA | 9.17 ± 0.00 bC | 9.09 ± 0.00 bC | 8.95 ± 0.07 bB | B2 | 6.27 ± 0.01 aA | 8.72 ± 0.00 bC | 9.09 ± 0.00 bC | 9.01 ± 0.03 bC |
| | A3 | 6.77 ± 0.10 aA | 6.83 ± 0.11 aB | 6.86 ± 0.01 aA | 5.90 ± 0.62 bA | B3 | 6.88 ± 0.04 aAB | 6.95 ± 0.08 aB | 7.10 ± 0.18 aB | 6.41 ± 0.01 abB |
| *Lactococcus lactis* sub. *cremoris* M78 | | | | | | | | | | |
| | A1 | 6.52 ± 0.02 aA | <7.00 aA | 7.57 ± 0.24 bA | 6.30 ± 0.00 aB | B1 | 6.59 ± 0.08 aA | 8.97 ± 0.16 cB | 9.19 ± 0.01 cB | 7.08 ± 0.32 abB |
| | A2 | 6.21 ± 0.03 cA | 8.06 ± 0.13 dB | 8.20 ± 0.03 dB | <4.00 aA | B2 | 6.22 ± 0.03 cA | 7.96 ± 0.04 dA | 7.98 ± 0.10 dA | 5.30 ± 0.00 bA |
| | A3 | 6.56 ± 0.06 aA | 9.19 ± 0.01 cC | 9.34 ± 0.01 cC | 7.82 ± 0.45 bC | B3 | 6.57 ± 0.00 aA | 9.18 ± 0.00 cB | 9.00 ± 0.02 cB | 6.71 ± 0.01 aB |
| *Lactiplantibacillus plantarum* H25 | | | | | | | | | | |
| | A1 | NA | NA | NA | NA | B1 | 5.63 ± 0.32 aA | 7.24 ± 0.06 bA | 7.30 ± 0.42 bA | 6.22 ± 0.37 aA |
| | A2 | NA | NA | NA | NA | B2 | 6.08 ± 0.21 aA | 7.08 ± 0.00 bA | 7.74 ± 0.06 cAB | 7.77 ± 0.05 cC |
| | A3 | NA | NA | NA | NA | B3 | 6.15 ± 0.08 aA | 8.92 ± 0.00 cB | 8.21 ± 0.13 cB | 6.98 ± 0.07 bB |

[1] Means ± standard deviations of three enumerations; within a row, means with different lowercase letters are significantly different (*p* < 0.05); within a column, separate means of each LAB strain with different uppercase letters are significantly different (*p* < 0.05). [2] 'Cheese milk', 'Acidified curd', 'Fresh RTE cheese', and 'Ripened cheese' are four sequential critical processing steps according to the artisanal Galotyri cheese processing protocol illustrated in Figure 1 and explained in Section 2.2. NA: not applicable; ST1 +M78: *S. thermophilus* ST1+*Lc. lactis* subsp. *cremoris* M78; ST1+M78+H25: *S. thermophilus* ST1+*Lc. lactis* subsp. *cremoris* M78+*L. plantarum* H25.

**Table 2.** Populations (log CFU/g) of autochthonous enterococci and non-LAB spoilage bacteria and yeast contaminating biota, selectively enumerated infour sequential processing steps of three independent batches of pilot-scale-produced, traditional Galotyri PDO cheese [1].

| Type of Microbial Contamination | | Galotyri Cheese A (Basic Starter Mix: ST1+M78) [2] | | | | | Galotyri Cheese B (Starter Mix: ST1+M78+H25) [2] | | | |
|---|---|---|---|---|---|---|---|---|---|---|
| | Trial/ Batch | Cheese Milk | Acidified Curd | Fresh RTE Cheese | Ripened Cheese | Trial/ Batch | Cheese Milk | Acidified Curd | Fresh RTE Cheese | Ripened Cheese |
| Enterococci | A1 | <2.00 aA | 5.09 ± 0.27 cC | 5.96 ± 0.28 dC | 5.48 ± 0.10 cdC | B1 | <2.00 aA | <2.00 aA | 4.04 ± 0.37 bB | 4.78 ± 0.10 cC |
| | A2 | <2.00 aA | <2.00 aA | <2.00 aA | <2.00 aA | B2 | <2.00 aA | <2.00 aA | <2.00 aA | <2.00 aA |
| | A3 | <2.00 aA | 3.77 ± 0.12 cB | 3.76 ± 0.04 cB | 2.90 ± 0.00 bB | B3 | <2.00 aA | <2.00 aA | <2.00 aA | 2.60 ± 0.00 bB |
| Coliform bacteria | A1 | <1.00 aA | 7.80 ± 0.04 dC | 6.25 ± 0.07 cC | 4.38 ± 0.00 bC | B1 | <1.00 aA | 4.25 ± 0.07 bB | 3.97 ± 0.15 bB | <1.00 aA |
| | A2 | <1.00 aA | <1.00 aA | <1.00 aA | <1.00 aA | B2 | <1.00 aA | <1.00 aA | <1.00 aA | <1.00 aA |
| | A3 | <1.00 aA | 3.31 ± 0.10 cB | 2.85 ± 0.35 bcB | 2.30 ± 0.01 bB | B3 | <1.00 aA | <1.00 aA | <1.00 aA | <1.00 aA |

**Table 2.** *Cont.*

| Type of Microbial Contamination | | Galotyri Cheese A (Basic Starter Mix: ST1+M78) [2] | | | | | Galotyri Cheese B (Starter Mix: ST1+M78+H25) [2] | | | |
|---|---|---|---|---|---|---|---|---|---|---|
| | Trial/ Batch | Cheese Milk | Acidified Curd | Fresh RTE Cheese | Ripened Cheese | Trial/ Batch | Cheese Milk | Acidified Curd | Fresh RTE Cheese | Ripened Cheese |
| Pseudomonad-like bacteria | A1 | <2.00 aA | 7.51 ± 0.10 cB | 7.05 ± 0.10 cB | 5.82 ± 0.05 * | B1 | <2.00 aA | 4.01 ± 0.15 bB | 3.63 ± 0.31 bB | 5.14 ± 0.10 * |
| | A2 | <2.00 aA | <2.00 aA | <2.00 aA | <2.00 aA | B2 | <2.00 aA | <2.00 aA | <2.00 aA | <2.00 aA |
| | A3 | <2.00 aA | <2.00 aA | <2.00 aA | <2.00 aA | B3 | <2.00 aA | <2.00 aA | <2.00 aA | <2.00 aA |
| Total staphylococci | A1 | <2.00 aA | 4.91 ± 0.08 cC | <2.00 aA | 2.54 ± 0.34 bB | B1 | <2.00 aA | <2.00 aA | <2.00 aA | <2.00 aA |
| | A2 | <2.00 aA | <2.00 aA | <2.00 aA | <2.00 aA | B2 | <2.00 aA | <2.00 aA | <2.00 aA | <2.00 aA |
| | A3 | <2.00 aA | 4.06 ± 0.04 cB | <2.00 aA | 2.47 ± 0.00 bB | B3 | <2.00 aA | 4.33 ± 0.01 cB | 2.70 ± 0.01 bB | <2.00 aA |
| Yeasts | A1 | <2.00 aA | <2.00 aA | 3.48 ± 0.00 cB | 6.40 ± 0.01 eA | B1 | <2.00 aA | <2.00 aA | <3.00 bA | 5.52 ± 0.13 dA |
| | A2 | <2.00 aA | <2.00 aA | 4.74 ± 0.17 bC | 7.98 ± 0.18 dB | B2 | <2.00 aA | <2.00 aA | 4.30 ± 0.01 bB | 6.92 ± 0.12 cC |
| | A3 | <2.00 aA | 2.60 ± 0.01 bB | <3.00 bA | 6.65 ± 0.02 dA | B3 | <2.00 aA | 3.28 ± 0.03 cB | <3.00 bA | 6.25 ± 0.02 dB |

[1] Means ± standard deviations of three enumerations; within a row, means with different lowercase letters are significantly different ($p < 0.05$); within a column, separate means of each contaminant microbial group with different uppercase letters are significantly different ($p < 0.05$). Both A1 values bearing an asterisk in the column 'Ripened cheese' were yeast populations that grew predominantly and fully outnumbered pseudomonad-like bacteria on the selective CFC/25C agar plates. [2] 'Cheese milk', 'Acidified curd', 'Fresh RTE cheese', and 'Ripened cheese' are four sequential critical processing steps according to the Galotyri cheese processing protocol illustrated in Figure 1 and described in Section 2.2. ST1+M78: *S. thermophilus* ST1+*Lc. lactis* subsp. *cremoris* M78; ST1+M78+H25: *S. thermophilus* ST1+*Lc. lactis* subsp. *cremoris* M78+*L. plantarum* H25.

### 3.2. pH Values and Gross Composition of the Artisanal Galotyri Cheeses

The mean pH values of the fresh RTE Galotyri A (pH 4.57) and B (pH 4.52) cheeses were very similar (Table 3), indicating that the growth of the *L. plantarum* H25 adjunct strain in the latter samples had no significant ($p > 0.05$) effects on cheese pH. The mean pH values of the ripened Galotyri A and B cheese samples increased slightly (0.03–0.04 pH units) compared to the fresh RTE cheese samples, indicating that the major yeast growth and potential proteolysis during ripening also had minor ($p > 0.05$) effects on pH (Table 3). Regarding the cheese trial effects, the pH values of the fresh Galotyri A2 and B2 samples predominantly fermented by *S. thermophilus* ST1 were the lowest (pH 4.33–4.35). Conversely, the most contaminated fresh RTE Galotyri A1 cheese had the highest pH (4.72), while the pH of the corresponding fresh Galotyri B1 cheese—dominated by the NisA+ strain M78— was 4.62. Minor changes ($p > 0.05$) in the pH values of each independent cheese trial or treatment occurred during ripening.

**Table 3.** Physicochemical characteristics of traditional Galotyri PDO cheese fermented with or without supplementation of the basic starter LAB culture with the adjunct strain *L. plantarum* H25 [1].

| Parameter | Galotyri Cheese A (Basic Starter Mix: ST1+M78) [2] | | Galotyri Cheese B (Starter Mix: ST1+M78+H25) [2] | |
|---|---|---|---|---|
| | Fresh RTE Cheese | Ripened Cheese | Fresh RTE Cheese | Ripened Cheese |
| pH | 4.57 ± 0.21 a | 4.61 ± 0.19 a | 4.52 ± 0.15 a | 4.55 ± 0.20 a |
| Moisture (%) | 66.32 ± 5.46 a | 65.59 ± 6.44 a | 67.81 ± 4.17 a | 66.59 ± 3.29 a |
| Fat (%) | 12.25 ± 0.66 ab | 13.00 ± 1.80 b | 11.88 ± 0.82 a | 12.25 ± 0.43 ab |
| Protein (%) | 12.15 ± 0.63 b | 11.77 ± 0.27 ab | 11.44 ± 1.50 ab | 10.62 ± 1.08 a |
| Salt (NaCl) (%) | 2.23 ± 0.80 a | 2.29 ± 0.92 a | 2.05 ± 0.49 a | 1.98 ± 0.52 a |
| Ash (%) | 2.85 ± 0.76 a | 2.73 ± 0.79 a | 2.51 ± 0.30 a | 2.51 ± 0.45 a |
| Acidity (%) | 1.19 ± 0.42 a | 1.25 ± 0.36 a | 1.18 ± 0.47 a | 1.27 ± 0.38 a |

[1] Values are the means of three independent cheese trials ($n = 3$); within a row, means with different letters are significantly different ($p < 0.05$). [2] ST1 +M78: *S. thermophilus* ST1+*Lc. lactis* subsp. *cremoris* M78; ST1+M78+H25: *S. thermophilus* ST1+*Lc. lactis* subsp. *cremoris* M78+*L. plantarum* H25.

Moisture content was not significantly ($p > 0.05$) different among the fresh cheeses when the corresponding data were pooled for all three trials (Table 3). However, the Galotyri A (ST+M78) samples displayed ca. 1.5% lower mean moisture contents than the Galotyri B (ST1+M78+H25) samples before ripening. This overall trend was retained after 30 days of cold ripening, indicating that most cheeses with the *L. plantarum* H25 adjunct strain held slightly more (1–1.5%) water than their respective control (ST1+M78) cheeses before and after ripening. The ripened cheeses became less moist than the respective fresh RTE cheeses (Table 3). However, this overall difference was not significant ($p > 0.05$), because major water loss was prevented by packaging all Galotyri cheese samples in water-impermeable plastic containers during ripening. In contrast, major differences in the water loss occurred during the preceding cutting, draining, and salting of the acidified curds, attributable to the poor monitoring of the artisanal manufacturing method in the commercial plant. This strong trial-dependent effect was reflected in the resultant fresh RTE cheeses. Particularly, the fresh A1 and B1 cheese samples of the first contaminated Galotyri trial were the most drained, as they had moisture contents of 60.03% and 63.04%, respectively, which were further reduced to 58.19% and 62.84% in the ripened A1 and B1 cheese samples, respectively. In contrast, the moisture contents of the trial A2/B2 and A3/B3 cheese samples ranged from ca. 69 to 71% and 68 to 70% before and after ripening, respectively (data not tabulated separately for each trial in Table 3).

No significant differences ($p > 0.05$) were observed between the mean fat (%) and protein (%) contents of the artisanal Galotyri A (ST1+M78) and B (ST1+M78+H25) cheeses during processing and ripening (Table 3). However, the fat and protein contents of the Galotyri B cheeses containing the *L. plantarum* H25 adjunct strain generally tended to be lower than those of the Galotyri A cheeses fermented with the basic starter only. Slight increases in fat content versus slight decreases in protein content occurred from the fresh to

the ripened A and B cheese samples in all trials (Table 3), probably because small protein (but not fat) fractions were released in the cheese whey drainage during ripening. Again the cheese trial effect strongly affected the fat and protein contents. Overall, the fresh A1 (ST1+M78) cheese of the first trial—the most contaminated and dried—had the highest fat content, which was 13% and increased further to 15% after ripening. Meanwhile, the fresh B1 (ST1+M78+H25) cheese had the lowest protein content, which was 9.89% and reduced further to 9.45% after ripening. All Galotyri cheese samples of the later two trials had intermediate fat and protein contents (data not tabulated separately).

After processing, the mean salt content of the fresh Galotyri A cheeses (2.23%) was higher than that of the fresh Galotyri B cheeses (2.05%), but this difference—also retained in the ripened A and B samples (2.29% vs. 1.98%)—was not significant (Table 3). However, major variations in the salt contents occurred among the three independent cheese trials, due to the instability of draining and dry salting of the acidified curds in the commercial plant. Indeed, while the fresh RTE cheese samples of the first, most dried trial were found to contain 2.99% (cheese A1) and 2.32% (cheese B1) salt, the fresh RTE A2 and B2 cheese samples processed in the DRD pilotplant contained only 1.39% and 1.48% salt, respectively. The salt content in the fresh RTE A3 and B3 cheese samples was reduced to 2.31% and 2.34%, respectively, after the draining and dry salting operations in the commercial plant were stabilized. All of the ripened cheeses showed slight (<0.1%) increases or decreases ($p > 0.05$) in their salt (%) contents compared to their fresh cheese counterparts, except for the A1 (ST1+M78) cheese, in which the salt content increased from 2.99% to 3.23% after ripening.

Variations in ash content among trials also occurred. Ash ranged from 2.28% (A2) to 3.71% (A1) and from 2.33% (B2) to 2.85% (B1) among the fresh Galotyri A and B cheeses, respectively. The mean ash content of the fresh Galotyri A cheeses was 2.85% and decreased slightly ($p > 0.05$) to 2.73% after ripening, while the fresh Galotyri B cheeses contained 2.51% ash, which remained unchanged ($p > 0.05$) in the ripened cheeses (Table 3).

Comparing the mean acidity values among the fresh Galotyri A (ST1+M78) and Galotyri B (ST1+M78+H25) samples, it can be observed that the *L. plantarum* H25 adjunct strain did not cause any significant increase or decrease in cheese acidity, expressed as % (*w/w*) lactic acid. This result was in accordance with the abovementioned minor effects of strain H25 on the pH of the fresh cheeses. Acidity showed a slight increasing trend ($p > 0.05$) in the ripened cheeses, associated with the continuing degradation of lactose and consequent organic acid production by LAB during Galotyri cold ripening, as described in the next section.

### 3.3. Sugar and Organic Acid Concentrations in the Artisanal Galotyri Cheeses

The mean lactose content ranged from 2.69 to 3.07 (g/100 g cheese) and from 2.44 to 2.55 (g/100 g cheese) among the fresh and the ripened Galotyri cheese samples, respectively (Table 4). Thus, (i) a considerable amount of lactose remained in the fresh RTE (drained and salted) cheese matrix (step 3) after fermentation for 24 h in ambient conditions (acidified curd; step 2); (ii) residual lactose decreased slightly ($p > 0.05$) during ripening, but it was not minimized in the ripened cheeses; and (iii) the *L. plantarum* H25 adjunct strain had slight additive effects on reducing residual lactose in the ripened Galotyri B cheeses. Conversely, the cheese trial effect was very much evident as regards the residual lactose content, which was the highest in the fresh RTE samples (3.68 to 4.17 g/100 g) and the ripened samples (3.58 to 3.13 g/100 g) of the A3/B3 trial conducted in the commercial plant in April. Residual lactose was ca. 1–2 g less in the earlier two winter trials, ranging from 2.17 to 2.88 (g/100 g cheese) and from 1.74 to 2.45 (g/100 g cheese) before and after ripening, respectively. Hence, the abundant total LAB growth in all acidified curds (step 2; Figure 1) before draining and salting took place at the expense of D-glucose which, thus, was undetectable in all fresh RTE and ripened Galotyri cheeses (Table 4). Conversely, D-galactose, also derived from the breakdown of lactose in milk, was at fairly constant low mean levels of ca. 0.5 (g/100 g cheese) in the fresh RTE cheeses, which reduced further

during ripening. The mean D-galactose reduction was greater ($p < 0.05$) in the Galotyri A (ST1+M78) than in the Galotyri B (ST1+M78+H25) ripened cheese samples (Table 4).

**Table 4.** Concentrations (g/100 g) of the main sugars and organic acids intraditional Galotyri PDO cheese fermented with or without supplementation of the basic starter LAB culture with the adjunct strain *L. plantarum* H25 [1].

| Biochemical Parameter | Galotyri Cheese A (Basic Starter Mix: ST1+M78) [2] | | Galotyri Cheese B (Starter Mix: ST1+M78+H25) [2] | |
|---|---|---|---|---|
| | Fresh RTE Cheese | Ripened RTE Cheese | Fresh RTE Cheese | Ripened Cheese |
| Lactose | 2.69 ± 0.86 a | 2.55 ± 0.90 a | 3.07 ± 1.01 a | 2.44 ± 0.70 a |
| D-glucose | n.d. | n.d | n.d. | n.d. |
| D-galactose | 0.54 ± 0.17 b | 0.25 ± 0.15 a | 0.55 ± 0.14 b | 0.43 ± 0.25 ab |
| Lactic acid | 2.47 ± 1.17 a | 2.50 ± 0.63 a | 2.62 ± 1.15 a | 2.34 ± 1.26 a |
| Citric acid | 0.22 ± 0.14 a | 0.21 ± 0.07 a | 0.27 ± 0.12 a | 0.23 ± 0.14 a |
| Acetic acid | n.d. | 1.04/n.d./n.d. | 1.68/n.d./n.d. | 1.63/n.d./n.d. |
| Propionic acid | 0.46/0.58/n.d. | 0.36/0.56/n.d. | 0.28/0.55/n.d. | 0.16/0.54/n.d. |

[1] Values are the means of three independent cheese trials ($n = 3$); within a row, means with different letters are significantly different ($p < 0.05$); n.d., Not detected. Means were not calculated for acetic and propionic acids, because one or more cheese trials had non-detectable (n.d.) levels; thus, the values for acetic and propionic acids separated by a slash refer to the concentrations in each trial (A1/B1, A2/B2, A3/B3) separately. [2] ST1 +M78: *S. thermophilus* ST1+*Lc. lactis* subsp. *cremoris* M78; ST1+M78+H25: *S. thermophilus* ST1+*Lc. lactis* subsp. *cremoris* M78+*L. plantarum* H25.

Lactic acid—the main organic acid formed by the breakdown of sugar during milk fermentation—was quantified at mean levels as high as 2.47 and 2.62 (g/100 g cheese) in the fresh RTE Galotyri A (ST1+M78) and Galotyri B (ST1+M78+H25) cheese samples, respectively (Table 4). No significant increases ($p > 0.05$) in the mean lactic acid contents of the cheeses occurred during ripening; instead, lactic acid tended to decrease slightly in most ripened Galotyri B samples containing the *L. plantarum* H25 adjunct strain (Table 4). Meanwhile, the cheese trial effect was quite strong again—the fresh A2 and B2 cheeses predominantly fermented (>9 log CFU/g) by *S. thermophilus* ST1 (Table 2) displayed the highest lactate (3.36 to 3.38 g/100 g cheese) contents, regardless of the presence of the H25 strain, but in correlation with their aforementioned lowest pH values of 4.3–4.4. In contrast, the fresh RTE samples of the third cheese trial, where the two mesophilic M78 and M78+H25 strains predominated (Table 2), displayed the lowest lactate contents (1.15 to 1.30 g/100 g cheese); the above major differences in lactate content were retained between the A2/B2 and A3/B3 Galotyri cheese samples after ripening.

The mean concentrations of citric acid were very similar among the fresh RTE cheese samples of all trials (0.22–0.27 g/100 g cheese), and they remained unchanged (0.21–0.23 g/100 g) during ripening (Table 4). Acetic acid was undetectable in all cheeses, with the exception of the fresh B1 and all ripened (A1/B1) samples of the first contaminated Galotyri cheese trial, which contained more than 1% acetic acid. Considerable levels (0.36 to 0.58 g/100 g cheese) of propionic acid were detected in the fresh RTE cheeses of the first two cheese trials, which were unchanged or reduced slightly in the ripened cheeses (Table 4).

*3.4. Proteolysis—Nitrogen Fractions of the Artisanal Galotyri Cheeses*

The values of the WSN, TCA-SN, and PTA-SN fractions of the fresh RTE and ripened Galotyri A (ST1+M78) and B (ST1+M78+H25) cheeses are presented as the % of total nitrogen (TN) in Table 5. In general, the expression of the nitrogen fraction as %TN normalizes the results, because the effect of moisture content is excluded. To better demonstrate and interpret variations in Galotyri proteolysis associated with the quite strong trial-dependent effects, the results in Table 5 are presented combined for all trials (i.e., as means ± SD values; $n = 3$), as well as separately for each trial; the SD values of the means within each trial are not shown, for table simplification. The corresponding REI, RDI, and FAAI indices, calculated as described in Section 2.5, are also presented in Table 5.

**Table 5.** Concentration (%) of total nitrogen and nitrogenous fractions of traditional Galotyri PDO cheese fermented with or without supplementation of the basic starter culture with the adjunct strain *L. plantarum* H25 [1].

| Protein Fraction | Galotyri Cheese A (Basic Starter Mix: ST1+M78) [2] | | | Galotyri Cheese B (Starter Mix: ST1+M78+H25) [2] | | |
|---|---|---|---|---|---|---|
| | Trial/Batch | Fresh RTE Cheese | Ripened Cheese | Trial/Batch | Fresh RTE Cheese | Ripened Cheese |
| Total nitrogen (TN%) | A1 | 2.01 | 1.89 | B1 | 1.55 | 1.48 |
| | A2 | 1.89 | 1.81 | B2 | 1.81 | 1.82 |
| | A3 | 1.81 | 1.84 | B3 | 2.02 | 2.01 |
| | Mean ± SD | 1.90 ± 0.10 [a] | 1.85 ± 0.04 [a] | Mean ± SD | 1.79 ± 0.24 [a] | 1.77 ± 0.27 [a] |
| WSN [a] (%TN) | A1 | 9.37 | 9.97 | B1 | 10.75 | 11.17 |
| | A2 | 9.15 | 7.66 | B2 | 6.93 | 6.23 |
| | A3 | 5.18 | 7.12 | B3 | 4.13 | 6.93 |
| | Mean ± SD | 7.90 ± 1.93 [a] | 8.25 ± 1.24 [a] | Mean ± SD | 7.27 ± 2.71 [a] | 8.11 ± 2.18 [a] |
| TCA [b] (%TN) | A1 | 8.57 | 9.15 | B1 | 10.84 | 11.07 |
| | A2 | 5.56 | 6.42 | B2 | 6.27 | 5.89 |
| | A3 | 4.32 | 5.89 | B3 | 5.69 | 6.38 |
| | Mean ± SD | 6.15 ± 1.78 [a] | 7.15 ± 1.43 [a] | Mean ± SD | 7.16 ± 2.3 [a] | 7.78 ± 2.33 [a] |
| PTA [c] (%TN) | A1 | 5.78 | 6.00 | B1 | 6.37 | 6.91 |
| | A2 | 4.11 | 4.18 | B2 | 4.22 | 4.89 |
| | A3 | 5.72 | 6.67 | B3 | 5.57 | 6.10 |
| | Mean ± SD | 5.20 ± 0.77 [a] | 5.62 ± 1.05 [a] | Mean ± SD | 5.39 ± 0.89 [a] | 5.97 ± 0.83 [a] |
| REI [d] | A1 | 4.66 | 5.27 | B1 | 6.94 | 7.55 |
| | A2 | 4.84 | 4.23 | B2 | 3.83 | 3.42 |
| | A3 | 2.86 | 3.87 | B3 | 2.04 | 3.45 |
| | Mean ± SD | 4.12 ± 0.89 [a] | 4.46 ± 0.59 [a] | Mean ± SD | 4.27 ± 2.02 [a] | 4.80 ± 1.94 [a] |
| RDI [e] | A1 | 4.27 | 4.84 | B1 | 6.99 | 7.48 |
| | A2 | 2.94 | 3.55 | B2 | 3.46 | 3.24 |
| | A3 | 3.16 | 3.20 | B3 | 2.82 | 3.17 |
| | Mean ± SD | 3.46 ± 0.58 [a] | 3.86 ± 0.71 [a] | Mean ± SD | 4.42 ± 1.83 [a] | 4.63 ± 2.02 [a] |
| FAAI [f] | A1 | 2.88 | 3.18 | B1 | 4.11 | 4.67 |
| | A2 | 2.18 | 2.31 | B2 | 2.33 | 2.69 |
| | A3 | 3.16 | 3.62 | B3 | 2.76 | 3.03 |
| | Mean ± SD | 2.74 ± 0.41 [a] | 3.04 ± 0.54 [a] | Mean ± SD | 3.07 ± 0.76 [a] | 3.46 ± 0.86 [a] |

[a] Water-soluble nitrogen, [b] 12% trichloroacetic-acid-soluble N, [c] 5% phosphotungstic-acid-soluble N, [d] REI: ripening extension index, [e] RDI: ripening depth index, [f] FAAI: free amino acid index. [1] Means ± standard deviations of three independent cheese trials; within a row, means with different letters are significantly different ($p < 0.05$). Means of two measurements are also given separately for each cheese trial (A1/B1, A2/B2, A3/B3); SD values of the means for each trial are not shown, for table simplification. [2] ST1 +M78: *S. thermophilus* ST1+*Lc. lactis* subsp. *cremoris* M78; ST1+M78+H25: *S. thermophilus* ST1+*Lc. lactis* subsp. *cremoris* M78+*L. plantarum* H25.

The TN% values of the Galotyri cheese samples, before or after ripening, ranged from 1.81% to 2.02%; thus, they were similar—except for the B1 cheese, which was characterized by the lowest TN%, whether fresh (1.55%) or ripened (1.48%). Overall, according to the analysis of variance, there were no statistically significant differences ($p > 0.05$) among the mean TN% values of the fresh RTE and the ripened cheeses. Furthermore, no statistically significant differences ($p > 0.05$) were found among the mean values of the WSN, TCA-SN, and PTA-SN fractions of the fresh RTE and the ripened cheeses, or among the Galotyri A (ST1+M78) and Galotyri B (ST1+M78+H25) cheeses before or after ripening (Table 5); the latter finding indicated that the *L. plantarum* H25 adjunct strain had minor effects on the proteolysis of the artisanal Galotyri cheeses, particularly during cold ripening for 30 days. In accordance, the corresponding mean REI, RDI, and FAAI indices were not statistically different (Table 5).

However, the mean WSN, TCA-SN, and PTA-SN values, along with their corresponding mean REI, RDI, and FAAI indices, increased slightly—but clearly—from the fresh RTE to the ripened cheese step of both the Galotyri A and B treatments; this was a constant trend

(Table 5). Hence, proteolysis progressed rather mildly in most Galotyri cheese samples during cold ripening, whereas the cheese trial effect was again stronger than the starter LAB or the ripening time effects and counteracted the significant differences between the means bearing high cheese-trial-dependent SD values (Table 5). Specifically, the nitrogen fraction values of the fresh RTE cheeses of the first, most contaminated and/or dried trial, were the highest—especially the WSN and TCA-SN values of the B1 samples, which increased to above 10% of TN. In contrast, the WSN and TCA-SN values of the fresh RTE cheeses of the third trial—predominantly fermented by the M78 starter strain—were the lowest. The above different trends in proteolysis were retained after ripening, when all of the WSN and TCA-SN values of the ripened A1/B1 and A3/B3 cheeses increased compared to their fresh cheese counterparts. In fact, decreases in WSN during ripening were observed only in the A2 (mainly) and B2 cheeses (Table 5). Moreover, it is noteworthy that the fresh A2 (mainly) and B2 cheeses—which were predominantly fermented by the thermophilic starter ST1—had fairly high WSN values, but the lowest PTA-SN values among all trials after processing and ripening. Within each trial, the REI, RDI, and FAAI indices followed the aforementioned increasing or decreasing trends of the WSN, TCA-SN, and PTA-SN values, respectively (Table 5).

*3.5. Sensory Evaluation of the Artisanal Galotyri Cheeses*

The scores of the organoleptic evaluation of the artisanal Galotyri A (ST1+M78) and B (ST1+M78+H25) cheeses after two weeks of ripening are shown in Table 6. Because the sensory quality attributes were more affected by the quite strong trial-dependent effects, the results in Table 6 are presented separately for the cheese products of the first (A1, B1) and the second (A2, B2) trials. Unfortunately, similar comparative sensory results for the third (A3, B3) cheese trial could not be obtained, because all five panelists were not physically present at work on the same sampling day for panel testing, due to the strict COVID-19 restrictions. No significant differences in the appearance and the texture of the cheeses were found, although the mean texture scores of the second trial were slightly higher compared to the first trial. Furthermore, the control A1 (ST1+M78) cheese of the first trial, which was the most contaminated and dried, had a dull, whitish, granular appearance that was not appreciated by most panelists. Moreover, this particular cheese received a much lower flavor score ($p < 0.05$) compared to cheese B1, and it clearly had the worst flavor and, thus, total quality scores ($p < 0.05$) among all of the cheeses (Table 6).Specifically, all panelists reported that cheese A1 had a very salty taste and a strong unpleasant sheep odor, which was not sensed in cheese B1. The products of the second trial were both of high quality, characterized by a mild aroma and a mildly acidic, sour, pleasant, and refreshing taste. Compared to cheese A2,which shared textural and flavor characteristics with traditional Greek strained yogurt, cheese B2 supplemented with the adjunct strain H25 had a more viscous body and a more piquant aroma and delicate taste that received slightly higher mean scores from the panelists (Table 6). Similarly, four of the five panelists judged that cheese B3 (ST1+M78+H25) had slightly better flavor characteristics than cheese A3 (ST1+M78; control), following our request to taste and simply rank the two batch products of the third trial at random times during ripening.

**Table 6.** Organoleptic characteristics of traditional Galotyri PDO cheese products after two weeks of ripening [1].

| Cheese Trial/Batch | Appearance (10) [2] | Body/Texture (40) | Flavor (50) | Total Quality (100) |
|---|---|---|---|---|
| A1 (ST1+M78) [3] | 8.0 ± 1.2 a | 33.6 ± 2.2 a | 29.0 ± 2.2 a | 70.6 ± 2.6 a |
| B1 (ST1+M78+H25) | 9.0 ± 0.7 a | 33.6 ± 1.7 a | 42.5 ± 2.5 b | 85.1 ± 4.6 b |
| A2 (ST1+M78) | 9.2 ± 0.4 a | 35.2 ± 1.8 a | 43.0 ± 4.5 b | 87.4 ± 5.1 b |
| B2 (ST1+M78+H25) | 9.2 ± 0.4 a | 36.5 ± 3.0 a | 44.8 ± 4.8 b | 90.5 ± 7.8 b |

[1] Values are given separately for each trial and are the means ± standard deviations of the five panelists' scores ($n = 5$); within a row, means with different letters are significantly different ($p < 0.05$). [2] Values in brackets show the maximum scores. [3] ST1 +M78: *S. thermophilus* ST1+*Lc. lactis* subsp. *cremoris* M78; ST1+M78+H25: *S. thermophilus* ST1+*Lc. lactis* subsp. *cremoris* M78+*L. plantarum* H25.

## 4. Discussion

Galotyri was selected as an ideal soft cheese model for studying the performance of novel Greek indigenous LAB strain mixtures as starter or bioprotective adjunct cultures [28], because it is the only Greek PDO cheese that is traditionally manufactured from boiled milk [20]. Moreover, although Galotyri has had PDO registration since 1994 [50], its manufacturing technology still varies greatly from region to region, as well as between dairy plants of the same region, e.g., Epirus or Thessaly [4,16]. Consequently, the microbial ecology and the total quality characteristics of commercial Galotyri PDO brand products sampled from the Greek market remain extremely variable [4,16,29,51]. For instance, Galotyri was the most diverse type amongst six Greek PDO cheese types according to bacterial populations, with three different product brands of this cheese being significantly different from one another in the non-metric multidimensional scaling (NMDS) plots of the bacterial OTUs assessed with amplicon metabarcoding [51]. Samelis and Kakouri [16,29] also concluded that Galotyri remains the most variably manufactured and microbiologically diverse traditional Greek PDO cheese. Likewise, the results of this study confirmed the occurrence of major biotechnological variability within the pilot-scale Galotyri cheese trials, despite all of them being processed by a standard artisanal manufacturing method.

Overall, the authentic Galotyri (or Galotiri) cheesemaking procedures, originally described by Zygouris [52] in 1952 and Anifantakis [12] in 1991, promote a rather slow fermentation and acidification of the boiled, salted (3–4%) milk at ambient (around 20 °C) temperatures for 2–3 days, even when back-sloping techniques are used [53,54]. Under commercial conditions, this artisanal fermentation method, which remains PDO-specified [20], would select for—and can be undertaken only by—mesophilic (i.e., starter) LAB species (mainly *Lc. Lactis*). In contrast, *S. thermophilus* and *Lb. delbrueckii* subsp. *bulgaricus,* which are symbiotic in natural or commercial yogurt starters used in many traditional Greek cheeses—including Feta cheese PDO [2,7,55]—grow poorly at ambient temperatures, while none of them grows in salt concentrations above 2%. Thus, they cannot be applied in Galotyri PDO production, unless the boiled or pasteurized milk is left to curdle unsalted at fermentation temperatures of at least 30 °C for 6 h [56], or preferably at 40–45 °C for shorter times, as with yogurt [29,30].

In recent years, pasteurization has replaced traditional boiling of the milk in nearly all commercial plants certified for Galotyri PDO production [16,29,51]; notably, the uncommon production of an artisanal Galotyri PDO cheese variety from raw (unpasteurized) sheep and goat milks in a cottage industry in Ioannina, Epirus, remains to date [4,51]. Moreover, natural starters have been replaced by CSCs in most Galotyri PDO productions, and the fermentation temperature programs and the salting procedures—i.e., salt is added to the curd after draining rather than to the milk before curdling—have also been modified [48,53]. In particular, although thermophilic yogurt-like LAB fermentations deviate from the authentic Galotyri cheese technology [12,20,52], they have been commonly applied for the commercial manufacture of this PDO cheese during the last 20 years—first at the industrial level [16,57], and more recently in small traditional dairies [29]. Therefore, Samelis and Kakouri [16] isolated *S. thermophilus* and *Lb. delbrueckii* subsp. *bulgaricus* strains as a

nearly pure symbiotic starter culture (97.4%) from all industrial fresh Galotyri PDO cheese samples collected from retail outlets in Ioannina in the years 2003–2006 and 2014–2019. Conversely, the artisan-type cheeses were ripened and more diversified in terms of LAB ecology, with an overall prevalence of *Lc. lactis* (19.8%), followed by *L. plantarum* (16.9%) and *S. thermophilus* (14.7%), whereas the total isolation frequency of *Lb. bulgaricus* (2.8%) was very low [16].

Similar results regarding the increased LAB species diversity of three artisan-type Galotyri PDO cheese varieties—all produced in SMEs located in Epirus and ripened for at least two months—were also reported by others [4,51]. In particular, based on the 16S rRNA identification of LAB isolates, *L. plantarum* prevailed [4], whereas *Lb. delbrueckii* subsp. *bulgaricus* was not identified per sein any of the above cheeses, whether culture-independently [51] or by LAB isolation procedures [4], suggesting that yogurt starters were not applied for Galotyri production in the above plants. In contrast, typical yogurt starter strains of *S. thermophilus* and *Lb. delbrueckii* were the most abundant LAB species in all retail batches of two other artisan-type Galotyri PDO cheese varieties produced in Epirus—one marketed fresh (Brand-K), and the other ripened (Brand-Z) [29]. Moreover, the fresh Brand-K cheeses were enriched in *L. plantarum*, while the ripened Brand-Z cheeses contained a more diverse LAB biota comprising *Lacticaseibacillusparacasei*, *Leuc. mesenteroides*, *Lentilacto-bacillus*, pediococci, and a few spontaneous nonstarter thermophilic streptococci, lactobacilli and enterococci. No *L. plantarum* or *Lc. lactis* strains were isolated from any of the ripened cheeses, since yogurt was the only natural starter used in the Galotyri-Z products. In contrast, a typical *Lc. lactis* subsp. *lactis* starter strain was sporadically isolated from the fresh cheeses, most probably originating—along with all *L. plantarum* isolates—from the Feta cheese granules dispersed in the fresh yogurt-like Galotyri-K cheese curd [29].

The above controversial microbial ecology data prompted us to match the artisanal technology [20] with current commercial needs to strengthen the declining authenticity of Galotyri PDO cheese products in the market. For this purpose, (i) no DVS CSCs were used, (ii) the basic starter ST1+M78 included single strains of *S. thermophilus* and *Lc. lactis* subsp. *cremoris* only, (iii) *L. plantarum* H25 was used as an adjunct culture to replace the dispersion of Feta cheese in the fresh Galotyri curds, (iv) no *Lb. delbrueckii* or natural yogurt starters were used, (v) the LAB-inoculated milk was left to curdle unsalted at 20–22 °C for 24 h to allow ST1 growth during the first 2–3 h post-inoculation, (vi) salting (1.5–2%) was carried outin the curd after draining at 12 °C for 3 days, and (vii) ripening was shortened to one month at 4 °C instead of at least two months at 8 °C. Overall, despite our efforts, it was difficult to keep the artisan cheesemaking protocol (Figure 1) constant in the commercial plant; thus, the performance of the basic starter ST1+M78 was variable between trials, contrary to our expectations based on the stable growth patterns of the ST1 and M78 strains in sterile raw milk (SRM) co-cultures under controlled fermentation conditions in our laboratory [26,27]. The fact that the primary acidifying starter *S. thermophilus* ST1 predominated only in the second cheese trial (Table 1), processed in the DRD pilot plant under strict supervision, indicated that the milk fermentation conditions during the processing of the first and third cheese trials in the commercial plant were unfavorable for ST1 growth. In particular, the time required for the cheese milk temperature to drop from ca. 40 °C after inoculation to ca. 35, 30, 25 °C, and finally equilibrate at 20 °C in the fermentation chamber was only measurable in the second DRD trial; it was ca. 1, 2, 3, and 6 h, respectively. It seems, therefore, that this milk cooling rate (around 30 °C for the first 3 h of fermentation) was slow enough to boost growth of the thermophilic ST1 starter. The respective milk cooling rates in the Pappas trials were probably sufficiently faster to suppress ST1 growth, but not the growth of the mesophilic starter strain *Lc. lactis* subsp. *cremoris* M78, which grew predominantly in the A3 and B3 cheeses of the third trial and in the B1 cheeses of the first trial (Table 1). The failure of M78 to predominate in the A1 cheeses was attributed to the major contamination and growth of Gram-negative bacteria (Table 2) and numerous antagonistic LAB contaminants from the dairy plant environment. Because the milk of all trials was found to contain from 0 to <10 CFU of total bacteria after boiling,

the high accidental cross-contamination of the A1 milk occurred post-thermally and was associated with improper cleaning of the plastic buckets used to ferment the cheese milk. Therefore, in the second and third trials, the plastic buckets were replaced with stainless steel ones, which were disinfected with steam after cleaning. In general, artisanal Galotyri is a 'sensitive' cheese, which is difficult to process properly if the milk is contaminated with spoilage bacteria after boiling and prior to the onset of the LAB fermentation [28,57]. Galotyri may also become unsafe if the milk is contaminated with pathogenic bacteria such as *L. monocytogenes*, *Staphylococcus aureus*, or *Escherichia coli* O157:H7 after boiling [53,58,59]. These previous challenge studies have shown that the above pathogens may survive but cannot grow if contamination occurs in the acidified curd or the final RTE acid-curd cheese. However, *L. monocytogenes* and *S. aureus* have the potential to grow and form SEs in Galotyri cheese milks at an early fermentation step—even in the presence of starter and bioprotective LAB strains [28,58].

In summary, altogether, the microbiological results showed that the artisanal Galotyri manufacturing method applied during this study does not ensure optimal growth of the mixed ST1+M78 starter with respect to the constant ability of the thermophilic strain ST1 to promote rapid and sufficient growth at ambient fermentation temperatures and perform as the primary milk acidifier—as *S. thermophilus* starters do in the milk of traditional Greek cheese types fermented at ≥35 °C for constant times [2,26,35]. In fact, the present Galotyri cheese trials were conducted from December to early April, when the commercial plant room was quite cold and the temperature in the commercial ripening chamber might fall below 20 °C to hinder ST1 growth. Nevertheless, upon failure of the ST1 strain to grow, the mesophilic NisA+ starter strain M78 predominated and became the primary milk acidifier in the B1 and A3/B3 Galotyri trials in replacement of ST1 (Table 1). On this basis, mixed thermophilic and mesophilic starter cultures containing *S. thermophilus* and *Lc. lactis* should preferably be used in artisan-type Galotyri PDO cheese fermentations. Alternatively, mesophilic starter cultures containing *Lc. lactis* are highly suitable for use.

The quality of Galotyri-type cheeses made with different starter cultures was first evaluated by Katsiari et al. [56], who used imported CSCs and compared the different cheeses by analyzing for compositional, lipolytic, and sensory characteristics, but not for microbiological characteristics. Four CSCs—two mesophilic (MA01 1, containing *Lc. lactis* subsp. *lactis* and *Lc. lactis* subsp. *cremoris*; and Probat 222, containing *Lc. lactis* subsp. *lactis*, *Lc. lactis* subsp. *cremoris*, *Lc. lactis* subsp. *lactis* biovar. *diacetylactis*, and *Leuc. mesenteroides* subsp. *cremoris*), one thermophilic (CH-1, containing *S. thermophilus* and *Lb. delbrueckii* subsp. *bulgaricus*), and one mixed (CHOOZIT MT 1, containing *Lc. lactis* subsp. *lactis*, *Lc. lactis* subsp. *cremoris*, *S. thermophilus*, and *Lb. delbrueckii* subsp. *bulgaricus*)—were used to make four cheese treatments by a standard method. The authors concluded that high-quality Galotyri could be produced by using any of those four CSCs, although the mesophilic MA01 1 culture produced cheese with the most consistent flavor and overall quality during storage. The cheese with the thermophilic CH-1 culture was the most acidic in terms of pH, titratable acidity, and sour taste, and it had a yogurt-like appearance and a less firm body than the other cheeses, similar to the A2 (ST1+M78) cheeses in this study. The cheese with the Probat 222 culture had the least liked flavor due to its high acetate content. The cheese with the mixed CHOOZIT MT1 culture was also well 'balanced' [56]. In an earlier study, Kondyli et al. [48] compared the chemical and sensory characteristics of Galotyri-type cheeses made with different procedures, but with constant use of the mesophilic CSC MA01 1. The cheese made with rennet and salting of the curd with 1.5% salt after draining (R+SC) had the most consistent quality during storage and was the most preferred by the panelists, compared to the dry-salted cheese made without rennet (SC) or the cheese made with salted milk and rennet (SM+R). In this study, all cheese trials were conducted with the R+SC procedure, and the results confirmed that Galotyri with high sensorial quality (Table 6) was produced when either the thermophilic ST1 (trial 2) or the mesophilic M78 (trial 3) starter strains predominated in the acidified curd; the sensory quality of trial 1—particularly of the A1 cheese—was low for the reasons discussed above.

Acidification at the appropriate rate and time is an essential feature of Galotyri cheese-making, since it has a major impact on the total quality, affecting the compositional and biochemical parameters as well as the microbial and sensory characteristics of the final RTE cheese [56]. According to its PDO specifications, Galotyri should have a maximum permitted moisture content of 75% and a minimum fat content in dry matter of 40%; minimum or maximum limits for additional compositional parameters (proteins, salt, etc.) or the pH are not specified [20]. The artisanal Galotyri cheese first described by Anifantakis [12] in 1991 had pH 3.9, moisture 70.8%, fat 13.8%, proteins 9.8%, and salt 2.76%. In comparison, the gross composition of the commercial Galotyri PDO cheeses reported by Danezis et al. [21] in 2020 was pH 3.83, moisture 74.0%, fat 10.9%, proteins 8.24%, and salt 1.17%. Thus, commercial Galotyri products today have similar pH values but higher moisture contents and, therefore, reduced fat, protein, and especially salt contents compared tothe authentic Galotyri, for profit reasons, as well as for reasons relating to the aforementioned technological modifications and consumer health concerns with respect to less fat and salt in cheese [16,48]. Contrary to the current trends, our artisanal cheesemaking protocol resulted in fresh RTE and ripened cheeses being closer to the authentic Galotyri [12] in terms of gross composition, except for their mean pH (4.6), which was higher, ranging from 4.3 (trial 2; ST1>M78) to 4.7 (trials 1 and 2; ST<M78). Similarly, Katsiari et al. [56] reported that the cheese made with the thermophilic CH-1 culture had a lower pH (4.1) than the other cheeses made with mesophilic CSCs (pH 4.44 to 4.59). Challenge Galotyri cheese trials conducted by others with the CH-1 CSC to validate the fate of enterotoxigenic *S. aureus* and SE production also had a final pH of 4.3, which did not change during storage [58]. In general, all commercial, industrial, or artisan-type Galotyri PDO cheese products from the Greek market analyzed from 2003 onwards had pH 3.7–4.0 [4,16,29,53], i.e., consistently lower than the pH 4.1–4.8 of all pilot-plant Galotyri cheeses produced for various experimental purposes [28,48,56,58]. Furthermore, after 2003, all commercial or pilot Galotyri cheese products had 74–75% moisture, with some of them exceeding the 75% maximum limit by 1–2% [48,53,56]. Moreover, all industrial or artisan-type Galotyri market cheeses analyzed by Rogga et al. [53] had fat and protein contents below 10% and 9%, respectively, whereas all brand-named artisan-type Galotyri PDO cheeses analyzed in 2014–2019 were found to contain 11.8–12.7% fat and 10.2–11.9% protein [16,29]. Finally, after 2003, the salt contents of all commercial or pilot (experimental) Galotyri cheeses were reduced to 1–2%, according to the analytical data or labeling information [16,29,48,53,56], except for the first pilot/industrial-type or pilot/artisan-type Galotyri cheeses produced in the DRD from boiled salted milk (SM) without rennet, according to an authentic procedure [50], which contained 2.9–3.1% salt [53].

Published data regarding glycolysis and proteolysis in retail Galotyri PDO cheeses do not exist, while those for pilot Galotyri cheeses are scarce. Kondyli et al. [48] found 3.2, 2.7, and 3.0% lactose in fresh (2-day-old) Galotyri-type cheeses made with the SM+R, SC, and R+SC procedures, respectively, and the mesophilic CSC MA011; our corresponding results in Table 4 are similar with the above previous findings. Lekkas et al. [59] found remarkably lower levels (0.6–1.5 g/100 g) of total lactate in commercial industrial or artisanal Galotyri cheese samples inoculated with *E. coli* O157:H7 and stored at 4 °C for 28 days. Conversely, similar levels of citrate and low (<0.05 g/100 g)-to-undetectable levels of acetate were found in the artisanal and industrial cheeses, respectively, before storage [59]; our corresponding citrate and acetate results in Table 4 also showed a similar trend. However, after storage, the artisanal cheeses studied by Lekkas et al. [59] contained up to 0.3 g/100 g of acetate—a level that was still 3 to 6 times lower than the acetate levels detected in the 'defective' A1/B1 cheese trial only before or after ripening (Table 4). Katsiari et al. [56] measured acetate as the lowest MW FFA acid and reported that the Galotyri C cheese made with the CSC Probat 222 had flavor defects, such as 'pungent' and 'rancid', and received the lowest flavor and overall quality scores because it had the highest acetate (42–45 mg/100 g) content. All cheeses made with the other three CSCs or with the preferred MA011 culture by different procedures had low acetate contents throughout storage [56,60]. Regarding

proteolysis, Kondyli et al. [48] determined the WSN (%TN) only at values ranging from 5.73 to 6.97 and 6.76 to 8.67 in Galotyri-type cheeses after 1 and 30 days of storage at 2–3 °C, respectively. In this study, only the 'defective' A1/B1 cheeses had higher WSN values, even fresh—probably because rennet [48], along with the outgrowth of Gram-negative bacteria, increased proteolysis.

In conclusion, the fresh RTE A2/B2 cheeses, with mean moisture 69.8%, fat 12.4%, protein 11.8%, salt 1.44%, and pH 4.3, represent a new, high-quality, artisan-type Galotyri PDO product fermented with the primary aid of *S. thermophilus* contained in the novel ST1+M78 and ST1+M78+H25 starter cultures. Conversely, the fresh RTE A3/B3 cheeses, with mean moisture 69.9%, fat 11.4%, protein 12.2%, salt 2.33%, and pH 4.62, are morehigh-quality artisanal Galotyri PDO cheeses subjected to typical mesophilic LAB fermentation by the indigenous *Lc. lactis* M78 starter and *L. plantarum* H25 adjunct strains, which grew compatibly at similar high (ca. 9 log CFU/g) levels in the acidified cheese curds (Table 1). Both fresh RTE cheese products were well drained before ripening; their sensory quality was improved during and after cold (4 °C) ripening for 30 days. The *L. plantarum* H25 adjunct strain had minor effects on the gross composition and the proteolysis of the B2/B3 cheeses, but it improved their texture and flavor, as well as their microbiological quality during processing and ripening. Particularly, spoilage yeast populations were reduced in all ripened Bcheeses (Table 2), probably due to the antifungal effects of strain H25. Selected *L. plantarum* strains are known to exert beneficial antifungal, antilisterial, probiotic, and aromatic activities in various cheese technologies [17,19,61–63].

This is the first report on the application of *L. plantarum* as an adjunct culture in Galotyri PDO cheese; consideration of the in situ activities of strain H25 was beyond the scope of this study. However, *L. plantarum* occurs naturally or is artificially enriched in Galotyri by the addition of Feta cheese [4,16,29], which is a rich source of *L. plantarum* (probiotic) strains [64–67]. Studies have been conducted on functional Greek Feta cheese and yogurt made with potential probiotic *L. plantarum* strains [14,68,69]; however, potential health benefits have yet to be elucidated in vivo [70]. Research on probiotics should be extended to Galotyri. Furthermore, the M78+H25—or similar novel NisA+ indigenous Greek starter/adjunct cultures—could replace the addition of commercial nisin and natamycin in Galotyri [71] and reduce the production cost. According to recent data published by ELGO-DIMITRA, the total Galotyri PDO cheese production for the year 2021 was only 326 tons, which is a very low quantity compared to the 120,147 tons of Feta PDO, 13,878 tons of organic Feta PDO, or even 870 tons of Katiki Domokou PDO, which has become first among all traditional Greek PDO soft acid-curd cheeses in terms of production [72]. Therefore, the authentic Galotyri PDO cheese must be preserved and protected. Additional, more advanced research studies and marketing strategies are required to restore the traditional manufacturing technology of Galotyri towards its authentication and market protection.

**Author Contributions:** Conceptualization, J.S. and D.P.; methodology, J.S., C.T. and L.B.; formal analysis, J.S., C.T., C.N., I.G. and A.K.; resources, J.S. and D.P.; data curation, J.S. and L.B.; writing—original draft preparation, J.S., C.T. and L.B.; writing—review and editing, J.S. and L.B.; project administration, J.S. All authors have read and agreed to the published version of the manuscript.

**Funding:** This research was funded by the European Union and Greek national funds through the EPAnEK 2014-2020 Operational Program for Competitiveness, Entrepreneurship, and Innovation, under RESEARCH-CREATE-INNOVATE (project: T1EDK-00968; project acronym BIO TRUST).

**Institutional Review Board Statement:** Not applicable.

**Informed Consent Statement:** Not applicable.

**Data Availability Statement:** Not applicable.

**Conflicts of Interest:** The authors declare no conflict of interest.

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
