# Peer review of "Pilot-Scale Production of Traditional Galotyri PDO Cheese from Boiled Ewes’ Milk Fermented with the Aid of Greek Indigenous Lactococcus lactis subsp. cremoris Starter and Lactiplantibacillus plantarum Adjunct Strains"

_fermentation, doi:10.3390/fermentation9040345_

Round 1

Reviewer 1 Report

1. The notation of starter cultures, should be standardized throughout the manuscript - once the authors write the full name and once the abbreviation

2. The ssp. notation should be corrected to subsp.

3. Authors must carefully check the paper according to Fermentation requirements, there is a lack of consistency in the notations, e.g. p >; Fig.2; double spaces, etc.

4. Line 139: reports explaining studies by other authors should be moved to the "Discussion" section

5. Line 155: add more details

6. Line 155: line 362 explain the meaning of "boiled milk"

7. Line 372: double „c” – please improve

8. Line 526: Why? Please explain

9. The Discussion section is only a review of reports by other authors. Reference to the results presented in the manuscript is missing. Should be corrected

Author Response

Dear Reviewer 1

Thanks for your report and positive evaluation. Please refer to the attached PDF file with our itemized responses.

Sincerely,

John Samelis 

Reviewer 2 Report

Article is very interesting and could be useful in order to standardize Galotyri cheese production. However, it could be interesting to compare results with traditional produced cheeses made with spontanous fermentation. Also, it could be interesting to check consumer perspective about it.

Author Response

Dear Reviewer 2

Thanks for your positive evaluation of our article. Attached please find a PDF file with our responses to your general comments. 

Sincerely,

John Samelis
